# Differences in Effects of Length-Dependent Regulation of Force and Ca^2+^ Transient in the Myocardial Trabeculae of the Rat Right Atrium and Ventricle

**DOI:** 10.3390/ijms24108960

**Published:** 2023-05-18

**Authors:** Oleg Lookin, Alexander Balakin, Yuri Protsenko

**Affiliations:** Institute of Immunology and Physiology, Ural Branch of Russian Academy of Sciences, 106 Pervomayskaya Str., 620049 Yekaterinburg, Russia; o.lookin@iip.uran.ru (O.L.); y.protsenko@iip.uran.ru (Y.P.)

**Keywords:** rat myocardium, right atrium, right ventricle, Frank-Starling mechanism, Ca^2+^ transient

## Abstract

The comparative differences in the fundamental mechanisms of contractility regulation and calcium handling of atrial and ventricular myocardium remain poorly studied. An isometric force–length protocol was performed for the entire range of preloads in isolated rat right atrial (RA) and ventricular (RV) trabeculae with simultaneous measurements of force (Frank-Starling mechanism) and Ca^2+^ transients (CaT). Differences were found between length-dependent effects in RA and RV muscles: (a) the RA muscles were stiffer, faster, and presented with weaker active force than the RV muscles throughout the preload range; (b) the active/passive force—length relationships were almost linear for the RA and RV muscles; (c) the value of the relative length-dependent growth of passive/active mechanical tension did not differ between the RA and RV muscles; (d) the time-to-peak and amplitude of CaT did not differ between the RA and RV muscles; (e) the CaT decay phase was essentially monotonic and almost independent of preload in the RA muscles, but not in the RV muscles. Higher peak tension, prolonged isometric twitch, and CaT in the RV muscle may be the result of higher Ca^2+^ buffering by myofilaments. The molecular mechanisms that constitute the Frank-Starling mechanism are common in the rat RA and RV myocardium.

## 1. Introduction

The assessment of atrial and ventricular contractility in isolation and during their interaction in the adaptation of myocardial structures to load conditions at all levels of organization is an important problem in the physiopathology of the heart. Indicators of myocardial contractility of the atria and ventricles are well coordinated and continuously adjusted during the cardiac cycle to maintain an optimal level of pumping function of a healthy heart, but they are disrupted in cardiac pathological conditions. It is unclear how effective heterometric regulation of atrial contraction is compared to ventricular contraction [1]. It has been demonstrated that the Frank–Starling mechanism is also operative in the atrium [2]. Atrial output increases as atrial diameter increases, which contributes to maintaining a normal stroke volume [3]. Moreover, the contractile function of the atrium may decrease with severe dilatation dilation [3]. It is unclear whether this may occur because the declining phase of the length–tension relationship is reached [4].

At the same time, fundamental differences in morphological and functional properties between the atrial and ventricular myocardium of mammals and humans are known [4,5,6,7,8,9,10]. The atrial cardiomyocyte is much thinner, and its volume is ~15% of the ventricular myocyte. This is consistent with the much lower relative density of the t-tubular network in small animal atrial cells [11,12,13,14] and the lack of transverse t-tubules in rats [15]. The lack of transverse t-tubules in atrial myocytes substantiates less synchronous Ca^2+^-induced Ca^2+^ release (CICR) from the sarcoplasmic reticulum (SR) into the cytosol and less uniform activation of myofilaments in these cells. Atrial myocytes have increased alternant activity and spontaneous Ca^2+^ release compared to ventricular myocytes [8,16,17]. Such a characteristic intracellular Ca^2+^ dynamic in the atrial myocytes is greatly dissimilar from the rapid and relatively uniform Ca^2+^ release from the SR in the ventricles [18,19]. In addition, the atrial myocardium reveals significantly higher expression levels and activity of the sarcoplasmic reticulum Ca^2+^-ATPase SERCA2a [6,8] and the high-velocity α-MHC myosin-ATPase [9]. Since the major and fast Ca^2+^ buffers in the cardiac cell are SERCA and the myofilament protein troponin C (TnC) [20,21,22,23], the atrioventricular differences in Ca^2+^ regulation may underlie the dissimilarity, not only in the kinetics of the systolic Ca^2+^ transient (CaT) [8,20,24], but also in the preload-dependent activation of contraction, which is greatly modulated by intracellular Ca^2+^ homeostasis. These differences provide the effectiveness of the pumping function.

The amplitude and time course of the contraction twitches are profoundly different in the atrial and ventricular myocardium. Tension developed by ventricular myocardium is greater (by about 30–50%), and the duration of the entire contraction–relaxation cycle is almost doubled in the ventricle compared to the atrium. These differences are only partly due to different myosin isoforms, as evident from force recordings in the mouse. In this species, ventricular and atrial myocardia express the same myosin isoform, i.e., >99% α in both chambers, but ventricular contraction is invariably stronger and slower (see, e.g., [4,25]. Thus, it was found that both length-dependent activation and length-dependent lattice reduction are clearly more pronounced in the bovine left ventricle than in the bovine left atrium, supporting the notion that titin regulates length-dependent activation by regulating the lattice spacing [26].

To date, no studies have focused directly on atrioventricular differences in preload-induced modulation of contractility and CaT dynamics. In the present study, we implemented an isometric force–length protocol for the entire range of preloads—from rest to optimal length—with simultaneous measurement of isometric force and Ca^2+^ transient, and we characterized their preload-induced changes in the rat right atrial and right ventricular muscles.

## 2. Results

The representative traces of steady-state isometric force twitches and CaT, obtained under different preloads in right atrial (RA) and right ventricular (RV) trabeculae of healthy rats, are shown in Figure 1A,B and Figure 1C,D, respectively. The traces represent a typical force–length (tension–length) protocol used to compare the extent of the Frank-Starling mechanism, as well as the influence of preload on CaT characteristics in rat atrial and ventricular myocardia. The principal differences between the preload-dependent behaviour of mechanical activity and CaT of rat RA and RV trabeculae are as follows: (1) the preload-induced increase in passive (end-diastolic) force values is considerably higher in RA muscles; (2) the rate of isometric force development and relaxation is substantially higher in RA, so the overall twitch duration is also substantially shorter; (3) the preload-dependent changes in the shape of CaT are much more expressed in RV compared to RA. For example, the increase in preload always induced a prominent short-term deceleration of the decay phase of CaT in the RV muscles, but not in the RA muscles (Figure 1D and Figure 1B, respectively). We will further evaluate this effect in detail.

The preload-dependent increase in peak (active) isometric tension was significantly higher in RV trabeculae vs. RA trabeculae (Figure 2A, *p* < 0.001 at any preload). In contrast, the value of end-diastolic passive tension was significantly higher in RA trabeculae at any preload, except slack length (Figure 2B). Interestingly, the relative preload-induced elevation of active or passive tension did not differ between the RA and RV muscles. Active tension increased by a factor of 1.55 ± 0.50 in RA and 1.51 ± 0.38 in RV per each 5% *L_opt_* of preload increment (mean ± SD, non-significant difference). For passive tension, the preload-related factors were 1.67 ± 0.52 and 1.67 ± 0.17, respectively, and they were not different between the groups.

The amplitude and end-diastolic level of CaT were not significantly different between muscles at any preload. Moreover, these characteristics displayed little or no preload-dependence (Figure 2C,D). Therefore, the inherent mechanisms for preload-dependent activation of contraction (the Frank-Starling mechanism) remained virtually the same in atrial and ventricular cardiac muscles. However, the absolute values of active and passive tension were highly dissimilar between these muscles, and the possible reasons for this are discussed in the relevant section.

The significance of maximal rates of rise and decay of mechanical tension, normalized to peak active tension, was significantly lower in the RV vs. RA muscles (Figure 3A). The normalized value of the rates always decreased with an increase in preload. It is important to note that the relationship between these normalized rates, if individual data points are pooled within the same type of muscle (i.e., RA or RV), was substantially linear, with a Spearman *r* value of ~0.78 (*p* < 0.0001) for both muscle sets. Furthermore, each individual relationship between normalized maximal rates of active tension rise and decay was approximated by a linear function, and the corresponding slope or intercept coefficients were pooled for all RA vs. RV muscles. It was found that the averaged values for the slope and intercept coefficients were not significantly different between the RA and RV muscles, e.g., the slope coefficient was 1.72 ± 0.65 and 1.39 ± 0.59, respectively (*p* = 0.44).

As the maximal rates of tension rise and decay were consistently lower in the RV muscles, these muscles showed systematically larger time-to-peak tension values, as well as the time values for decay from peak to 50% of amplitude (TTP and T50, respectively, in Figure 3B). Again, the Spearman *r* values for these time-related parameters were significantly high (0.63 and 0.84 for RA and RV muscles, respectively (*p* < 0.0001 for both muscle sets)), indicating a highly linear relationship between parameters of force development and relaxation. Thus, we conclude that the relative activation/inactivation rates of myofilaments, which are indirectly transformed to the maximal rates of active tension development (TTP) and relaxation (T50), remain highly similar in the atrial and ventricular myocardia of rats.

In contrast to this congruous interrelation of tension-related parameters between the RA and RV muscles, the same characteristics for CaT were highly discordant and, therefore, did not follow the same common relationship (Figure 3C,D). Moreover, neither CaT maximal rates of rise and decay (normalized to CaT peak), nor CaT time-to-peak and time of decay from peak to 50% of amplitude, were functionally related to each other. In fact, the characteristics related to the rising phase of CaT, i.e., maximal rate of rise and time-to-peak, were preload-dependent, while the characteristics of the CaT decay phase were insensitive to the changes in preload. This is clearly demonstrated by the nearly zero-slope character of all plots shown in Figure 3C,D. It is, therefore, conceivable that the mechanisms of transformation of Ca^2+^ activation of myofilaments to the contractile response may differ between the atrial and ventricular myocardia of rats.

The overall shape of CaT decay, as shown in Figure 1, is remarkably different between the RA and RV muscles. First, the rate of CaT decay is significantly higher in the RA muscles at any preload (see Figure 3C). Second, the discrepancies between the kinetics of CaT decay are especially prominent at moderate and high preloads. For example, CaT decays for exemplary RA and RV trabeculae in Figure 1 are nearly monotonic at a preload of 75% *L_opt_* (no stretch of muscles), but their behavior under gradual increase in preload is principally different. The increase in preload did not modify significantly the overall monotonic shape of the CaT decay in the RA muscles, but it was accompanied by substantial, although short-term, deceleration of this decay in the RV muscles, making it remarkably non-monotonous.

In our previous study, we have developed the “difference curves” method to estimate the full-phase preload-dependent changes in CaT ([27] for details). Briefly, the CaT trace measured at minimal preload (75% *L*_opt_) without the diastolic level of fluorescence and normalized to its maximum was used as the referent trace, making all its values range between 0 and 1. The same manipulation was applied to all other CaT traces, measured at the incrementally increased preloads. The offset-adjusted and self-normalized trace was then sequentially subtracted in a time-consistent manner from all other offset-adjusted and self-normalized CaT traces. Thus, the contribution of length changes to different phases of the CaT traces was determined by directly estimating the magnitude of the modulus value of the span of its deviations, i.e., the full amplitude (shown here in % of the CaT amplitude) and the total area under these deviations or the integral value (shown in % of the entire area under the CaT trace). Note the substantially higher effect of preload on the magnitude of difference curves in the RV trabecula when compared to the RA muscle. The examples of these difference curves for one RA and one RV trabeculae are shown in Figure 4A,B (the calculations were obtained from the records presented in Figure 1B,D).

Both the relative amplitude and the integral magnitude of the difference curves were significantly higher in the RV muscles compared to the RA muscles (*p* < 0.05) (Figure 4C,D). For example, the relative amplitude of the CaT difference curve increased from 2.3 ± 0.8% to 7.4 ± 2.4% in the RA trabeculae (~three-fold increase) and from 7.3 ± 3.3% to 40.5 ± 11.1% in the RV trabeculae (>three-fold increase) upon the increase in preload from 75% to 100% *L_opt_*. The integral magnitude of the CaT difference curve, in turn, increased from 1.6 ± 0.5% to 6.4 ± 2.5% in the RA trabeculae (~four-fold increase) and from 4.0 ± 1.7% to 24.8 ± 6.1% in the RV trabeculae (>six-fold increase).

According to the last point, we compared mutual changes in the peak active tension and the integral magnitude of the CaT difference curves (Figure 5). As expected, we found a direct relationship between the absolute value of peak active tension and the integral magnitude of CaT difference curve. Typically, the values for the RV muscles were an order of magnitude higher than those for the RA muscles (Figure 5A). In addition, we plotted the same characteristics as a % of the value measured at *L_opt_* (Figure 5B). Here, the relationships for the RA and RV muscles, in fact, followed the same line; Spearman *r* values for both relationships were >0.8, indicating their highly linear behaviour. This result indicates that the molecular mechanisms of Ca-induced myofilamental activation, i.e., Ca^2+^-dependent activation and inactivation of myofilaments are essentially similar between atrial and ventricular rat myocardia.

## 3. Discussion

### 3.1. RA vs. RV: The Preload-Induced Effects on the Isometric Twitch Characteristics and Frank-Starling Mechanism

This study revealed substantial differences in the characteristics of isometric contraction in the isolated right atrial and right ventricular trabeculae of healthy rats. The atrial muscles were much stiffer, but weaker and faster in the active isometric contraction, as compared to the ventricular muscles. In general, these findings are consistent with previously reported data [4,10,18,28,29,30]. However, the timing characteristics of mechanical relaxation were found to be highly affected by preload only in the ventricular myocardium, while the atrial muscles showed minor preload-induced changes. It is very likely that the two major features of the atrial muscles vs. ventricular muscles—much low active tension and the lack of sensitivity of time-to-peak and relaxation time to preload—may originate from the very prominent discrepancies found with regard to Ca^2+^ transient decay.

There are only a few reports in which the effect of preload on isometric twitch, the Frank-Starling mechanism, has been directly compared between mammalian atrial and ventricular myocardium (either in vitro or in vivo) [2,31,32]. In the range of physiological preloads (from 75% to 100% *L_opt_*), a linear relationship is shown between the value of isometric tension and preload of isolated trabeculae of the myocardium of the atrium and ventricle of rats (Figure 2). This range of preloads was chosen to avoid the effects of myocardial damage when the length corresponding to *L*_max_ is reached or slightly exceeded, and data comparison was convenient. A similar type was found for the pressure–volume relationship in the whole heart of dogs and the Frank-Starling mechanism law for ventricular wall segments and single cardiomyocytes in the articles [33,34].

The relative increase in active tension per single increase in the value of preload was the same in the trabeculae of the atria and ventricles (Figure 2A). This finding suggests that the intracellular processes as a “myofilament length-dependent activation” (LDA), underlying the Frank-Starling mechanism, are quite similar between the muscles, and they are influenced by preload in a similar manner [35]. This allows both chambers to interact synergistically during the cardiac cycle as the load changes. On the other hand, the lower atrial force in absolute values could be due to the lower concentration of free calcium in atrial cardiomyocytes compared to ventricular ones (Figure 2).

### 3.2. RA vs. RV: The Preload-Induced Effects on the Characteristics of Ca^2+^ Transient and Frank-Starling Mechanism

In the human atrial myocardium, preload has been shown to affect both contractility and diastolic/systolic cytosolic Ca^2+^ levels [32], while, in isolated atrial cardiomyocytes, no considerable effect of preload on Ca^2+^ transient characteristics was observed [36]. Some other mammals showed weaker effects of preload compared to ventricular myocardium [37]. This may be explained by the prominent differences in the expression of ion channels, regulatory and contractile proteins of myofilaments, and by more subtle differences in the structure of the sarcoplasmic membrane between ventricular and atrial myocytes [4,7,9,10]. Ventricular myocytes contain many crests that greatly facilitate mechanical and electrical communication between laterally adjacent myocytes [38]. In atrial myocytes, they are not expressed or lacked, which may reduce the mechanical synchronization of adjacent cells [38]. As a result, the relative contractile ability, i.e., force per cell cross-section, is lower in atrial muscles, as shown in previous studies [28,32,37] and in the present study.

The characteristics of CaT were divided into two main groups for analysis. The first group contains those characteristics which were shown to be very similar (either quantitatively or in the relation of their preload-dependent behavior) between the atrial and ventricular muscles: CaT amplitude and time-to-peak. The near preload-insensitivity of these characteristics showed that the initial pre-load of a muscle hardly affects both the amount of Ca^2+^ released from the sarcoplasmic reticulum (SR) and the kinetics of the rapid elevation of cytosolic Ca^2+^. Another group includes CaT relaxation time, total duration, and maximal rates of rise/decay, and the latter, in contrast, showed great discrepancy between the atrial and ventricular muscles.

The CaT signal in RA cardiomyocytes was found to increase monotonically with increasing preload, and the shape of this signal remained insensitive to preload (see, for example, Figure 1B). In contrast, cardiomyocytes of the RV showed a very strong preload-induced modulation of the CaT decay, which was monotonous in its shape only at low preloads and became non-monotonic at higher preloads (see Figure 1D). The non-monotonic configuration of CaT decay has been shown previously in rat ventricular myocardium and is known as a “bump” [39,40], but, for the first time, we assessed its deficit in the atrial myocardium. Our previous studies of this phenomenon have shown that the molecular basis of the “bump” were the two-stage kinetics of the formation-decay reaction of CaTn complexes, which are disrupted in the damaged myocardium, which leads to a decrease in the appearance of the “bump” [27,40]. Its absence in the atrium is most likely due to the high rate of Ca^2+^ sequestration into SR and the low buffering capacity of contractile proteins [6,8,28,29,41].

In this study, we have shown that the degree of utilization of Ca^2+^ by myofilaments, estimated by the relative amplitude and area of the curve of the time difference of Ca^2+^ at different preload, was several times greater in the ventricular muscles than in the atrial muscles. This could be the basis for the differences we found in CaT dynamics between atrial and ventricular myocytes. Some of our results for rat myocardia are consistent with previous studies performed on healthy atrial and ventricular human myocardium [4,42]. For example, some researchers have reported that the maximal rate of CaT is higher in human atrial myocytes, so time-to-peak occurs earlier than in human ventricular cells.

It can be concluded that the rat right atrial myocardium indeed demonstrates a reduced level of Ca^2+^ utilization by myofilaments, which differs from that of the ventricle, as indicated by a much greater increase in the characteristics of the curve of Ca^2+^ time difference caused by preload in the ventricle. However, a group of the main molecular mechanisms of regulation of myocardial contractility underlying the Frank-Starling mechanism, triggered by the overlap of myofilaments, length-dependent activation, etc., remains largely similar in the myocardium of the right atrium and right ventricle of healthy rats, which allows them to react cooperatively to changes in load to maintain the pumping function of the heart.

## 4. Materials and Methods

### 4.1. Ethical Approval

The animals involved in the present study were cared for according to Directive 2010/63/EU of the European Parliament and the Guide for the Care and Use of Laboratory Animals, published by the US National Institutes of Health (NIH Publication No. 85-23, revised 1985), and their use was approved by the local Institutional Ethics Committee (Protocol No. 14/20 of 8 December 2020). Male Wistar rats (*n* = 10, aged 1.5–2 month) were obtained from the Institutional Animal House and maintained under standard conditions (12h light/dark cycle, water and food ad libitum). The animals were rapidly euthanized by cervical dislocation immediately before the tissue collection.

### 4.2. Muscle Preparations

The heart was removed immediately after euthanasia and placed in a modified Krebs-Henseleit (KHB) solution in mM: NaCl 118, KCl 4.7, MgSO_4_ 1.2, KH_2_PO_4_ 1.2, NaHCO_3_ 25, CaCl_2_ 2, glucose 11.1, insulin 5U, pH adjusted to 7.35 at 25 °C under bubbling with 95% O_2_/5% CO_2_, containing 30 mM 2,3-butanedione monoxime (BDM). A right atrial (RA) or right ventricular (RV) trabecula (100 to 500 µm thick, 1–3 mm length) was dissected from the corresponding heart chamber and quickly attached to force transducer and length servomotor in the experimental bath under continuous perfusion by BDM-free KHB (saturated with 95%O_2_/5%CO_2_). For measurement of CaT, a trabecular muscle was pre-incubated in the saline with 5 μM fura-2/AM + 0.4% *w/v* Pluronic F-127. All measurements were conducted in the individual muscles (*n* = 10 for RA strips and *n* = 9 for RV trabeculae), which were paced by electrical impulses at 2 Hz and 30 °C. All chemicals were from Sigma-Aldrich (St. Louis, MO, USA).

### 4.3. Data Acquisition

The force value of isometric contraction at different lengths, and fura-2 fluorescence signal of the trabecular muscle, were measured simultaneously with the Muscle Research System (Scientific Instruments, Heidelberg, Germany) and sampled at 10 kHz via DAC/ADC (PCI-1716S; AdLink Technology, New Taipei City, Taiwan) using our custom-made software running in a real-time environment (HyperKernel; Nematron, Ann Arbor, MI, USA), integrated with Windows XP (Microsoft; Redmond, WA, USA). CaT was obtained as the ratio of fluorescence signals obtained at 510 nm, when excited at 340/380 nm (*F* = *F_340_*/*F_380_*) and presented here as *F*/*F_0_*, where *F_0_* is a ratio measured in quiescent non-stretched muscle. Measurements were made under steady-state conditions in a series of preloads of 75 (minimal active tension), 80, 85, 90, 95, and 100% *L_opt_*. The muscles were stretched by 25% of the initial length in order to neutralize mechanical damage, and this preload is referred to as ‘optimal’ *L_opt_* near *L*_max_ [43].

### 4.4. Force-Length Protocol

A trabecular muscle was fully released to produce zero passive force and then gradually stretched every time by 50 µm steps and allowed to equilibrate at each new length (preload) prior to the measurement of its active/passive force and CaT. The ‘optimal’ preload (*L_opt_*) was achieved when a decrease in the amplitude of the active tension was observed with a further increase in the length of the trabeculae.

The series of isometric contractions and CaT obtained at different preloads were used to construct the tension–preload relationship, as well as all preload-related relationships for the CaT characteristics. The mechanical tension values were obtained from the force values by normalizing them to the muscle cross-section. The cross-section was calculated, assuming the elliptical formula *S* = *πd*^2^/12, where *d* is the larger width of the muscle measured in the transversal direction.

### 4.5. Statistical Analysis

Statistical analyses were carried out with GraphPad Prism 7.0 (GraphPad Software, San Diego, CA, USA). The Mann-Whitney U-test was used to evaluate the significance of the difference between the mean values of the isometric tension/CaT characteristics for RA and RV muscles measured under the same preload. The differences were considered significant at *p* < 0.05, *p* < 0.01 and *p* < 0.001. Data are presented as mean ± SD.

## 5. Conclusions

The morpho-functional performances of the right atrial myocardium are essentially different from those of the right ventricular myocardium in rats, allowing them to interact effectively during the cardiac cycle. Based on our results, we hypothesize that the extent of Ca^2+^ buffering by myofilaments is several times higher in ventricular vs. atrial muscle, resulting in higher peak tension, prolonged isometric twitch, and Ca^2+^ transient with remarkable non-monotonic decay. However, the molecular mechanisms, which constitute the Frank-Starling mechanism, are common in the rat atrial and ventricular myocardium, making the preload-dependent relative changes in contractility essentially the same in these muscles.

## Figures and Tables

**Figure 1 ijms-24-08960-f001:**
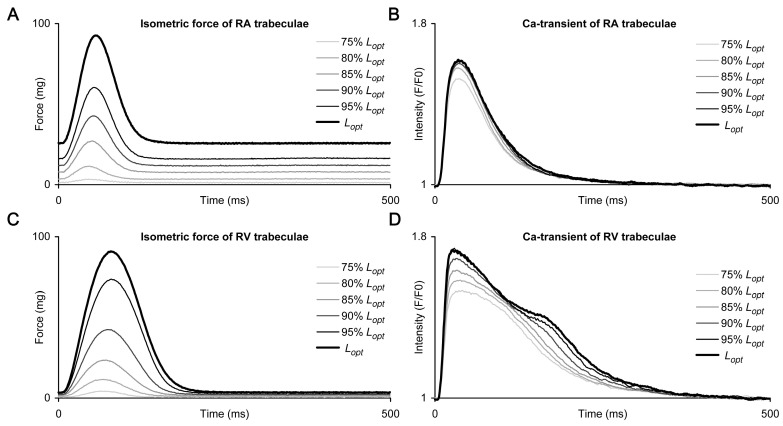
The representative traces of steady-state isometric force (**A**,**C**) and Ca^2+^ transients (**B**,**D**) obtained under different preloads in right atrial (RA, top panels) and right ventricular (RV, bottom panels) trabeculae of healthy rats. The preload values are indicated in the legends. Muscles were paced by electrical impulses at a frequency of 2 Hz and a temperature of the KHB solution 30 °C.

**Figure 2 ijms-24-08960-f002:**
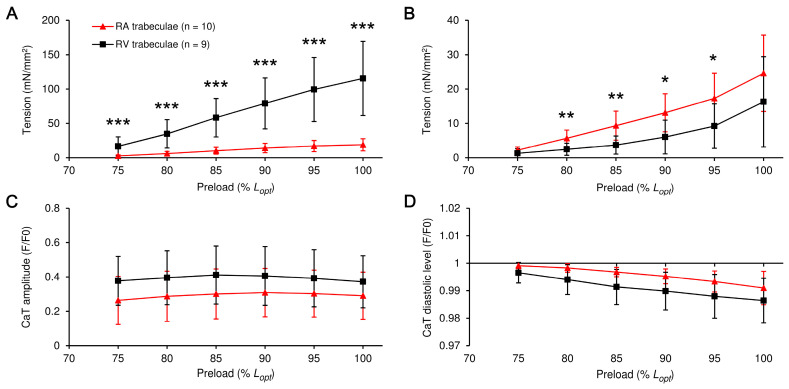
The preload-dependent effect on the amplitude of active tension (**A**) and Ca^2+^ transient, CaT, (**C**) or diastolic (passive) tension (**B**) and end-diastolic level of CaT (**D**) in right atrial (RA) and right ventricular (RV) trabeculae from healthy rats. CaT parameters are given in F/F0 units, where F0 is the fluorescence in non-stimulated non-stretched muscle. The number of muscles is given in the legend in (**A**), and this legend is valid for all panels. The difference between RA and RV for the given preload is significant at * *p* < 0.05, ** *p* < 0.01, or *** *p* < 0.001 (Mann-Whitney U-test). All data presented are means ± SDs.

**Figure 3 ijms-24-08960-f003:**
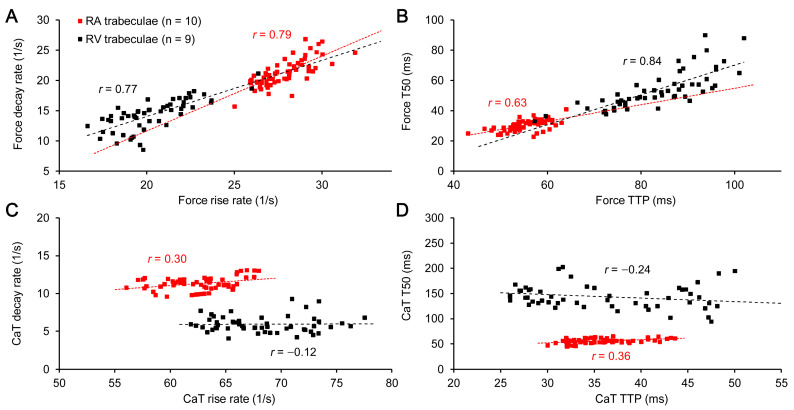
The relationships between maximal rate of rise and maximal rate of relaxation/decay (**A**,**C**) and time-to-peak, TTP, and time of relaxation from peak to 50% amplitude, T50, (**B**,**D**), for active tension (**A**,**B**) and decay for Ca^2+^ transient, CaT, (**C,D**) in the atrial (RA) and right ventricular (RV) trabeculae of healthy rats. The values of maximal rise and relaxation rates of tension (**A**) or decay for CaT (**C**) are normalized to the corresponding amplitude of the signal. The number of muscles used to construct the plots is given in the legend in (**A**), and this legend is valid for all panels; the number of individual points is a product of total muscle number and the number of tested preloads (in total = 6, from 75% *L_opt_* to 100% *L_opt_* with 5% *L_opt_* incremental step). The Spearman r values are shown separately for each data cloud.

**Figure 4 ijms-24-08960-f004:**
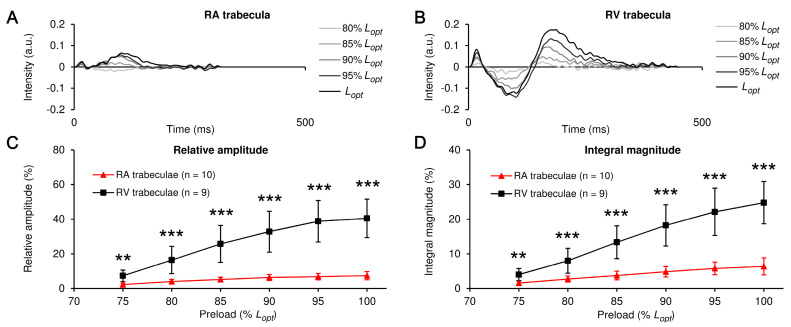
The implementation of the method of Ca^2+^ transients difference curves for indirect evaluation of the extent of Ca–TnC interaction in the right atrial (RA) and right ventricular (RV) myocardia of rats. The exemplary traces of the CaT difference curves obtained under different preloads for RA and RV muscles (**A**,**B**). The preload values are shown in the legends as a % of *L_opt_*. The averaged preload-dependent effects on the relative amplitude (**C**) and integral magnitude (**D**) of the difference curves obtained for RA or RV muscles (**C**,**D**). The number of muscles is indicated in the legends. The difference between RA and RV for the given preload is significant at ** *p* < 0.01 or *** *p* < 0.001 (Mann-Whitney U-test). Data in (**C**,**D**) are presented as means ± SDs.

**Figure 5 ijms-24-08960-f005:**
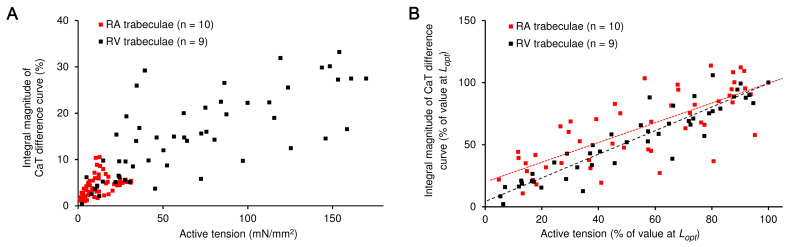
The preload-dependent relationships between peak active tension and integral magnitude of Ca^2+^ transient (CaT) difference curve in the atrial (RA) and right ventricular (RV) trabeculae of rats. (**A**) The relationships are plotted for the absolute values of the characteristics as they changed with the preload. (**B**) The relationships are plotted for the characteristics normalized to the value measured at the maximal preload *L_opt_*. The number of muscles used to construct the plots is indicated in the legends. The Spearman *r* values are shown in (**B**), separately, for each data cloud.

## Data Availability

Data generated or analyzed during this study are available from the corresponding author upon reasonable request.

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
