# Peer review of "Differences in Effects of Length-Dependent Regulation of Force and Ca2+ Transient in the Myocardial Trabeculae of the Rat Right Atrium and Ventricle"

_ijms, 2023, doi:10.3390/ijms24108960_

Round 1

Reviewer 1 Report

The authors have presented an exciting manuscript comparing the differences in contractility regulation and calcium handling of ventricular and atrial myocytes. The manuscript will be of interest to the readers of IJMS. But some minor issues need to be addressed before accepting this paper.

General comments:

First, I would like to say the paper is well-structured and well-presented.

Authors should provide representative images of time laps calcium imaging in the manuscript.

Not required, but some crucial controls could be included for the conclusions made on the calcium transient. Control samples treated with caffeine to deplete the SR calcium reserves and thapsigargin to block SERCA. This will strengthen the conclusion there is a difference in calcium utilization between atria and ventricles, not a difference in calcium release from the SR. Without these controls, it is hard to support the conclusion about calcium utilization. And this possibility should be discussed in the discussion.

The English of the article is good.

Author Response

General comments:

First, I would like to say the paper is well-structured and well-presented.

We thank the reviewer for the positive evaluation of our study.

Point 1: Authors should provide representative images of time laps calcium imaging in the manuscript.

Response 1: The representative traces of the steady-state isometric Ca2+ transients obtained for right atrial and ventricular muscles under different preloads are already shown in Figure 1(B, D).

Figure 1. The representative traces of steady-state isometric force (A, C) and Ca2+ transients (B, D) obtained under different preloads in right atrial (RA, top panels) and right ventricular (RV, bottom panels) trabeculae of healthy rats. The preload values are indicated in the legends. Muscles were paced by electrical impulses at a frequency of 2 Hz and a temperature of the KHB solution 30 °C.

Point 2: Not required, but some crucial controls could be included for the conclusions made on the calcium transient. Control samples treated with caffeine to deplete the SR calcium reserves and thapsigargin to block SERCA. This will strengthen the conclusion there is a difference in calcium utilization between atria and ventricles, not a difference in calcium release from the SR. Without these controls, it is hard to support the conclusion about calcium utilization. And this possibility should be discussed in the discussion.

Response 2: Indeed, the results of the experiments with caffeine and thapsigargin may strengthen our conclusions. However, there is a problem in the implementation of such experiments. The main aim of our work was to compare the effects and mechanisms of heterometric (i.e. length-dependent) regulation of contractility and Ca2+ transient between healthy atrial vs ventricular myocardium (in other words, to compare the extents of the Frank-Starling Mechanism). Treatment with caffeine or thapsigargin leads to an inability to reconstruct the length-force relationship of a muscle. Of course, the proposed approach can be done at any fixed preload (muscle length) in a separate series of experiments, but this was not the aim of our present study.

Reviewer 2 Report

In the manuscript title “Differences in effects of length-dependent regulation of force and Ca2+ transient in the myocardial trabeculae of the rat right atrium and ventricle”, the Authors show that extent of Ca2+ buffering by myofilaments is higher in ventricular when compared with atrial cardiac muscle, resulting in higher peak tension, prolonged isometric twitch and Ca2+ transient with remarkable non-monotonic decay.

I found the manuscript interesting and well written, moreover, authors’ conclusion are adequate with respect to results. Nevertheless, I have some comments:

1) “Materials and methos” section should to be moved after the “Introduction” section, before “Results”. In this section Authors should indicate haw many rats they used, as well how many cells were analyzed.

2) Please, changes the following words as follow:

- Line 28: “physiology and pathology” change in “physiopathology”

- Line 31: “cardiac pathology” change “in cardiac pathological conditions”

3) I would like the authors postulate the role of contraction and Ca2+ transient difference between atrial and ventricle cells, with respect to cardiac diseases (as well as atrial fibrillation, heart failure etc.). If you need, you can cite the following paper “The gut hormone ghrelin partially reverses energy substrate metabolic alterations in the failing heart. Circ Heart Fail. 2014 Jul;7(4):643-51. doi: 10.1161/CIRCHEARTFAILURE.114.001167” where calcium transients and fractional shortening were analyzed in ventricle healthy dog cells and compared with heart failure dog cells. 

Minor English change are required (see comments 2)

Author Response

I found the manuscript interesting and well written, moreover, authors’ conclusion are adequate with respect to results.

We thank the reviewer for the positive evaluation of our study.

Nevertheless, I have some comments:

Point 1: “Materials and methods” section should to be moved after the “Introduction” section, before “Results”. In this section Authors should indicate haw many rats they used, as well how many cells were analyzed.

Response 1: The article was prepared according to "Instructions for Authors".

Research manuscript sections: Introduction, Results, Discussion, Materials and Methods, Conclusions (optional).” (https://www.mdpi.com/journal/ijms/instructions)

The number of rats has been added to the "4.1. Ethical approval" section.

“Male Wistar rats (n=10, aged 1.5-2 month) were obtained from the Institutional Animal House and maintained under standard conditions (12h light/dark cycle, water and food ad libitum). The animals were rapidly euthanized by cervical dislocation immediately before the tissue collection.”

The number of muscles has been added to the section "4.2. Muscle preparations".

“All measurements were conducted in the individual muscles (n = 10 for RA strips and n = 9 for RV trabeculae), which were paced by electrical impulses at 2 Hz and 30 °C.”

Point 2: Please, changes the following words as follow:

- Line 28: “physiology and pathology” change in “physiopathology”

- Line 31: “cardiac pathology” change “in cardiac pathological conditions”

Response 2: Two replacements have been made in the "1. Introduction" section.

Point 3: I would like the authors postulate the role of contraction and Ca2+ transient difference between atrial and ventricle cells, with respect to cardiac diseases (as well as atrial fibrillation, heart failure etc.). If you need, you can cite the following paper “The gut hormone ghrelin partially reverses energy substrate metabolic alterations in the failing heart. Circ Heart Fail. 2014 Jul;7(4):643-51. doi: 10.1161/CIRCHEARTFAILURE.114.001167” where calcium transients and fractional shortening were analyzed in ventricle healthy dog cells and compared with heart failure dog cells.

Response 3: Thank you for your interesting remark and the proposed paper.

However, the main aim of our study was to compare the effects and mechanisms of heterometric regulation of isometric contraction and Ca2+ transient of healthy atrial and ventricular myocardium. There are no published data about such length-dependent effects in a pathologically altered myocardium. This is the aim for our next article. Also, there are a limited number of reports comparing Ca2+ transients in healthy atrial and ventricular myocardium with the same length of cardiomyocytes, mainly in animals. Therefore, we don’t think that the proposed paper (Circ Heart Fail. 2014 Jul; 7(4):643-51. doi: 10.1161/CIRCHEARTFAILURE.114.001167) really fits well to the present study.

In our article, we have already mentioned some previously published data about the significant differences in the parameters of a single twitch (mechanical stress and Ca2+ transient) between the atrial and ventricular myocardium:

“Ca2+ uptake into ventricular SR vesicles was increased by 13% in the presence of protein kinase A while that into atrial SR vesicles remained unaffected. Western blotting analysis revealed 23% less SERCA2a protein, but 76% more PLB in ventricular compared to atrial tissue” (from Freestone et al. 2000, reference [29]).

“Atrial cells had smaller systolic Ca2+ transients that decayed more rapidly. This was due primarily to an increased rate of SR mediated Ca2+ uptake. SR Ca2+ content was 289% greater and Ca2+ buffering capacity was increased ∼3-fold in atrial cells (from Walden et al. 2009, reference [8]).

The authors of the following articles paid their attention to the strengthening of the role of the atrium in the regulation of ventricular ejection in the development of pathology. “A recent review has convincingly shown that intracellular Ca2+ transients in atrial cardiomyocytes are markedly heterogeneous and fundamentally different from ventricular cardiomyocytes” (from Brandenburg et al. 2016, article is not included in the list of references). “In 51 patients with PH (48 with pulmonary arterial hypertension), RA conduit function was assessed as echocardiographic and was found that altered RV lusitropy is associated with impaired RA conduit phase. PEDSR (peak early diastolic strain rate) emerged as a promising, non-invasive, bedside-ready parameter to evaluate RV diastolic function and to predict prognosis in PH” (from Richter et al. 2020, article is not included in the list of references).

Nevertheless, we thank for the interesting remark. Of course, based on our own and published data, we could assume that the shape of Ca2+ transients in pathology will change significantly compared to the control ones. However, at present there is a lack of the comparison between length-dependent effects in the ventricular and atrial myocardium.
